

# Enhancing e-learning through AI: advanced techniques for optimizing student performance

Rund Mahafdah[1], Seifeddine Bouallegue[2] and Ridha Bouallegue[1]

[1] Innov'COM Laboratory High School of Communications (Sup'COM), University of Carthage, Carthage, Tunisia
[2] University of Doha for Science and Technology, Doha, Qatar

## ABSTRACT

The integration of Artificial Intelligence (AI) into e-learning has transformed conventional educational approaches, improving the learning process and maximizing student achievement. This study offers a thorough examination of how AI can be utilized to enhance e-learning results by employing advanced predictive methods and performance optimization strategies. The main goals consist of creating an AI-based framework to monitor and analyze student interactions, evaluating the influence of online learning platforms on student understanding using advanced algorithms, and determining the most efficient methods for blended learning systems. AI algorithms, known for their cognitive ability and capacity to learn, adapt, and make decisions, are employed to analyze and forecast student performance, thereby improving educational quality and outcomes. The practical results obtained by implementing machine learning and deep learning models, such as convolutional neural networks (CNN) and recurrent neural networks (RNN), show substantial enhancements in forecasting different performance metrics. This research highlights the ability of AI to develop adaptable, effective, and successful e-learning environments, promoting enhanced academic achievement and customized learning experiences. The findings demonstrate that CNN outperformed other deep learning and machine learning algorithms in terms of accuracy during the prediction phase, showcasing the advanced capabilities of AI in educational contexts. Portions of this text were previously published as part of a preprint (https://doi.org/10.21203/rs.3.rs-4724603/v1).

# INTRODUCTION

The development and growth of any nation depend entirely upon the education system followed by the country. There has been a significant change in the teaching methodology and systems during the past five decades, unlike the older age. Many improvements have been observed and identified with various techniques integrated with the learning methodology. Researchers have proposed various models for improvement in teaching and learning methods. The old strategy for teaching and learning at the last level has required upgrades and improvements for a long time (*Nodir Djamolovhich, 2019*).

Corresponding author
Rund Mahafdah,
rund.mahafdah@supcom.tn

The mindset of the students towards that school and college was expecting a reformation. Many surveys are done with the view in mind of working towards the improvement of the education system. Many authors proposed that various modern approaches would help with the required improvement. The implementation and integration of these modern technical ideas should be done with the help of technology (*Jabbarova, 2020*). The elements of the education system that are expected to be improved are categorized as follows:

- Classroom teaching: Widely acknowledged as the most conventional pedagogical approach, classroom teaching has long been a fundamental means of delivering education for successive generations. This study affirms that classroom instruction continues to be one of the most participatory and efficient means of disseminating knowledge (*Jabbarova, 2020*).

- Mobile learning: Mobile learning, also known as self-directed learning, is gaining popularity among young people and has become an important educational method. A significant number of pupils choose this approach because of its adaptability and ease of use (*Lin et al., 2019*).

- The use of learning management systems (LMSs) with diverse functionalities is becoming increasingly popular among students and teachers. Several firms offer licensed and open-source LMS systems, emphasizing the significance of adequate training for optimal utilization. Research suggests that the use of (LMS) is widespread in higher education, especially in the Middle East (*Khan & Qudrat-Ullah, 2020*).

- Virtual classrooms: This cutting-edge educational approach enables teachers to establish remote connections with students. It enables easy access to knowledge and learning, promoting interactive communication between teachers and students through integrated technology.

- Blended learning: Blended learning is the integration of several teaching approaches, such as the use of gadgets, LMS systems, virtual tutoring, and mobile learning, to produce a full educational experience (*Alnahdi, 2019*).

- Smart classroom: Smart classrooms employ technological gadgets and equipment to improve student involvement and acceptance (*Rojabi, 2020*). Organizations are creating intelligent classroom tools to promote interaction and sustain student engagement, utilizing smart boards and panels as significant educational resources while reducing the need for human interaction (*Rojabi, 2020*).

The education industry has seen a substantial transition towards digital learning settings, particularly due to global events that need remote education solutions. Although there are many e-learning platforms available, it is still difficult to ensure that students perform at their best owing to problems such as lack of engagement, limited accessibility, and the need for personalized learning experiences. AI can improve e-learning by using predictive analytics to detect and assist students who are at risk, creating personalized learning routes, and implementing real-time feedback systems. The current challenges e-learning systems face include a need for personalized learning paths and suboptimal

student engagement. These issues hinder the optimization of student outcomes through traditional AI models. Nevertheless, the use of AI in e-learning is now in its early phases, and its complete capacity to enhance educational results has not yet been completely achieved. As e-learning platforms expand quickly, the problem of delivering scalable, efficient, and customized education has gained prominence. One-size-fits-all approaches are frequently used by traditional e-learning systems, which ignore the differences in each student's demands, learning preferences, aptitudes, and performance. As a result, there are high dropout rates and unequal educational outcomes since many students find it difficult to stay interested, remember information, or perform at their best. Additionally, especially when working with large groups of students, it can be challenging for instructors and institutions to keep track of students' progress and provide timely feedback. These difficulties reduce the general efficacy and efficiency of e-learning systems. The primary problem is the lack of a thorough framework for effectively integrating AI into e-learning platforms so that student performance may be reliably predicted and enhanced. Current e-learning platforms frequently fail to make efficient use of the massive amounts of data generated in order to offer instructors and students useful insights. Thus, research and development in AI technologies are desperately needed in order to produce predictive models that have the potential to significantly improve student performance, retention, and engagement.

This research aims to develop AI-based models to enhance personalized learning recommendations, strengthen student engagement through adaptive content delivery, and optimize student performance prediction. The main goals are to use IoT to track student interactions in Middle Eastern educational environments, assess the influence of online learning platforms on student understanding using sophisticated algorithms, and determine the most effective approaches for blended learning systems. In addition, artificial intelligence algorithms are used to analyze and predict student performance. The questions to be answered through this research work are:

RQ1: Is it possible to improve student performance and solve the problem of their lack of participation accessibility?

RQ2: How to integrate artificial intelligence with e-learning systems and improve its performance?

Our research work is structured as follows: "Introduction": the related works that show the previous approaches to deal with this problem are discussed in "Literature Review". In "Methodology", the methodology that was utilized is outlined in this section. The utilization of the dataset, the application of preprocessing methods, and the implementation of artificial intelligence techniques, specifically machine learning and deep learning, are the three primary steps that comprise this methodology. The results of the proposed showcase the outcomes of utilizing machine learning algorithms (random forest (RF), decision tree (DT), extreme gradient boost (XGB), k-neural network (KNN)) and deep learning models (CNN, RNN, long short term network (LSTM), artificial neural network (ANN)) to forecast five distinct class labels: Performance Level, Final Grade, Adaptivity Level, Happy and Sad, and Focused and Unfocused. are discussed in "Experimental Results". "Discussion" includes the conclusions reached in this article.

Finally, "Conclusion" concludes the article and reports suggested possible future works. Portions of this text were previously published as part of a preprint (https://doi.org/10.21203/rs.3.rs-4724603/v1).

## LITERATURE REVIEW

This section describes summary of previous works that are related to this article in three sections: Associative (*Abuhammad, 2020*), cognitive (*Leach & Walker, 2000*; *Wahyudi, Agustin & Ambarawati, 2022*), and situational theories (*Manjoo, 2016*).

### Distant learning with cognitive skills

There are many university distance learning systems. Some of these systems emphasize adult learning more than school-level student modules (*Leach & Walker, 2000*). Some learning mechanisms prioritize theoretical concepts over pedagogical models (*Wahyudi, Agustin & Ambarawati, 2022*). Associative, cognitive, and situational theories have been reviewed in distant learning situations. The first is activity-based learning. However, the second emphasizes learner abilities.

The third one only addresses learning from certain situations. The three should increase many learner personality byproducts. Improved thinking power, exclusive learning, reasoning, and a positive, real-time perspective are supposed to determine system eligibility. Some distant learning models emphasize technology (*Abdul et al., 2022*). Many of them target management, development, accessibility, and environmental factors.

### Frameworks used in distant e-learning

The success of e-learning platforms is heavily reliant on the motivational factors of both learners and mentors, as there are numerous platforms available. The mentor's capacity to articulate the system in an inspiring manner is vital. Multiple authors have presented diverse frameworks throughout history. *Abuhammad (2020)* proposed a learning framework that utilizes multimedia to increase student motivation. Similarly, the authors in *Goudeau et al. (2021)* suggested utilizing an objective learning-based approach in conjunction with constructive mind development. These frameworks have demonstrated numerous benefits based on research.

*Ferri, Grifoni & Guzzo (2020)* were able to create a framework that addressed online learning using R2D2 (Read, Reflect, Display, and Do). This is because of the unique character of online and electronic learning environments which emphasize graphic, video, and activity, as opposed to verbal or minimalist, explanation of the content. Such as virtual travels, auditory lectures or video-based explanations, and animations. Apart from this, practical skills oriented content is enhanced by various tasks calling for analytical thinking. However, it needed the improvement of data collection methods in order to be considered totally successful.

The implementation of online learning faced some resistance in one of the education systems such as China. Many teachers were uncomfortable working under new teaching modalities, so were a majority of the students who received detailed and comprehensive

study materials. In order to tackle these problems, *Tang et al. (2020)* implemented an intelligent framework that utilizes an automated questioning system to actively involve students and offer them answers. This approach successfully identified student learning patterns. Data analysis was utilized to enhance learning and teaching outcomes through the application of diverse data mining techniques.

## Social media and sentiment analysis

Social media is reportedly one of the most popular ways to spread knowledge today. All people, including businesspeople and government officials, use social media to connect and express their opinions. Twitter, Facebook, Instagram, and others are used to find businesses in various fields using technology. *Manjoo (2016)* found that social media was used to endorse or oppose election candidates. In Saudi Arabia, young people prefer social media.

Integration of social media platforms with e-learning can solve many collaborative, communication-based, and problem-specific activities, according to *Troussas, Krouska & Sgouropoulou (2021)*. Student familiarity with media platforms makes it easy for them to adapt learning methods with social media-based e-learning systems. This type of social media can mine student behavior and information to identify cognitive improvement.

Sentiment analysis with social media analysis can reveal a person's mindset on specific topics. Sentiment analysis based on social media posts can be done using many computational algorithms. Special application programming interfaces (APIs) and natural language processing engines can identify student sentiment and prospects.

*Zhang et al. (2020a)* used social media platforms to identify posts, public opinions, blogs, tweets, and other psychometric instruments to classify them as anger, happiness, depression, confusion, or tension. Multilingual social media platforms are a major cause of user misinformation. Since the complete transition from one language to another is not easy, a minimal linguistic computational solution may allow for variation in results.

Multiple languages make sentimental analysis of mass opinions difficult. Sentiment analysis and social media give students a powerful platform to learn and improve. Today, the global platform relies on social media, and youth are greatly impacted by it. This study recommends using such platforms to measure student sentiment as a model parameter.

## Use of IoT in learning

With the rise of IoT devices in industries and homes, a large population now relies on them. Successful device integration provides anytime learning solutions. Many mature and practical IOT applications are deployed worldwide. The popularity and availability of cheaper circuits are driving IoT device use. IoT-enabled environments now have cheaper sensors, actuators, and other devices. Tablets and smartphones are affordable.

Additionally, Internet access at the fingertips is affordable. Internet access and low device prices make it easier to integrate IoT devices with e-learning systems. According to *Mircea, Stoica & Ghilic-Micu (2021)*, IoT services in e-learning offer many benefits. Remote laboratories are a key use of IoT-enabled services during the creation of e-learning

environments. Urban laboratories in ruble areas that cannot be upgraded can use remote laboratories to work and learn locally. Due to low student numbers, rural labs are not economically viable.

The combination of networking and communication technologies made many people's dreams come true by expanding educational facilities and systems with advanced teaching materials at affordable rates, which are available as online and remote learning resources among others. With these facilities, one can study using laboratory equipment such as a computer that is connected to the internet. Cloud computing, Internet of Things (IoT), and mobile computing have also made learning, especially education, advanced. In conjunction with the above, such tools and technologies as: Web cameras, RFID, Arduino, Raspberry Pie, and actuators play a big role in remote training.

New studies have added insight into associative, cognitive and situational learning theories. For instance, the associative learning has been explored using word-object co-occurrence models. On the other hand, cognitive theories have emphasized emotional and social interaction in digital environments. The situational learning research has also shown that collaborative learning impacts student engagement and interest.

## Activity and analysis of behavior

An understanding of the student's remote environment can be one of the important factors that will be responsible for identifying the knowledge delivery mechanism. Distance learning systems do not focus on the behavioral analysis of a student, which is done with the assistance of IoT-enabled devices. The eligibility to understand. The activity of an individual is one of the most challenging tasks in the distant learning paradigms. However, once it is achieved, it can be used to deliver the lessons in a very adaptive situation and a very simple manner for all the scheduled students in a class.

The analysis of the activities that the students are engaged in is yet another technique for the identification of the work characteristics of the individual. The body language, as well as the movement of the student, can be identified, observed, and analyzed with the help of various techniques. As per the study given by *Ericsson (2016)*, Reading the heartbeat of a student with the help of a smartwatch during an online exam can be one of the important tasks that will provide the student feedback based on his input. At the time of answering the heartbeat records, he will be able to identify his mindset and personality to recognize the situation on a macro level. The behavior of the student can be identified with the help of this reading to propose nervousness as well as stress, which in return can suggest the pattern of answers given by an individual.

## The use of artificial intelligence

Machine learning has become increasingly significant in recent times, alongside deep learning techniques, to identify patterns and trends across various domains worldwide. According to *Hämäläinen, Laine & Sutinen (2006)*, basic applications can possess significant capabilities. Using web access and online data processing, the inputs are analyzed. To achieve better results, it is possible to advocate for utilization of different deep learning algorithms, which can make a e-learning system more

effective over classroom learning. It is therefore a good practice to use the well-known methods of the naïve Bayesian, support vector machines along with the k-nearest neighbor method and this can as well result to accuracy of the results. This is not the case in the number of deep learning algorithms mainly designed to improve data mining that exist at present.

## METHODOLOGY

This section explains the method or methods used in this study to help reach specific objectives like prediction. The methodology for this study will be done in three steps namely the utilization of the data set, outlining pre-processing practices, and employing artificial intelligence such as machine learning and deep learning. Figure 1 shows the methodology step-by-step.

Design science research (DSR) methodology

A study was conducted in this instance applying design science research (DSR) methodology to come up with and evaluate an AI-driven framework for optimizing student performance in e-learning settings. This specific approach, known as DSR methodology, takes a systematic course through these central activities:

1. Problem definition:

The most significant challenges that are faced by e-learning systems are lack of personalized learning paths, less efficient student engagement and poor performance monitoring. Consequently, students' performance never attains optimal learning outcomes. Specifically, this study uses state-of-the-art AI techniques to enhance student performance prediction, engagement strategies among other areas.

2. Solution objectives:

Specifically the aim is to develop an AI-based framework with the following attributes:

- Enhancement in the personalization of learning recommendations grounded on student performance and behavior
- Incremental student's participation through adaptive content delivery
- Refinements on the performance prediction models for better learning experienced with RF, CNN and another form of machine learning algorithm
- Its success will be gauged on basis of prediction accuracy in terms of performance indicators, metrics measuring engagement as well as how computationally intensive such a system might become.

3. Design and development:

An e-learning system with an integrated AI-based predictive model was created as part of these studies. This model combines advanced techniques such as RF and CNN to forecast academic results from scratch hence coming up with personalized study paths for each learner based on historical data analysis. We included various aspects related to student's performance like interaction patterns, quiz outcomes, and studying habits for us to increase precision in making forecasts.

4. Demonstration:

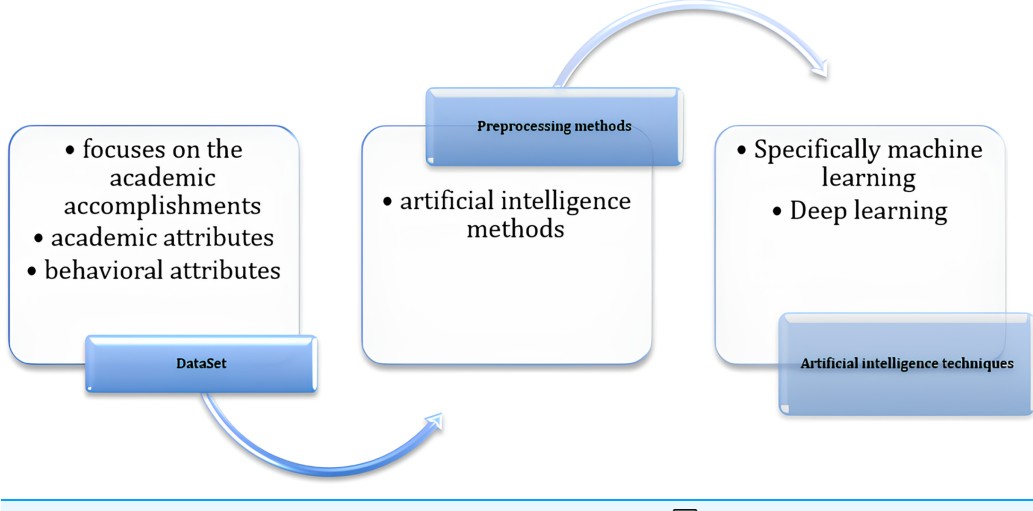

**Figure 1  The methodology.**               

We applied the framework that developed to real-world data set gathered from different Middle Eastern educational settings involving multiple e-learning interactions of students in them. The framework revealed actionable insights to administrators regarding students' learning trends and gave customized recommendations for improving student experience.

5. Evaluation:

The evaluation metrics for assessing the artifact was provided as follows:

- Accuracy prediction of learner outcomes
- Increased engagement by means of personalized learning trajectory
- Computational efficiency which involves time taken during computation and resource usage

It was found out that accuracy of student performance forecasts could be enhanced without compromising much on computation time. This evaluation showed significant advances in predicting student outcomes and maintaining a balance between precision and computational expense. However, the study noted few limitations like the fact that the model could not generalize so well with other datasets. Therefore, these areas deserve more attention from follow-up researches.

6. Communication:

Thus, the role of AI-driven techniques in revolutionizing the learning process as well as outcomes is exemplified. These findings provide insights on how academic researchers and educators can leverage data science solutions to improve students' academic performance.

## Dataset description

We utilized a dataset that was constructed by merging three publicly available datasets related to university student performance:

Dataset 1: Student Performance Prediction Dataset
Dataset 2: Students' Adaptability Level in Online Education
Dataset 3: xAPI-Edu-Data

We subjected the Kaggle dataset on education records of college students from the Middle East to a study and downloaded the file from the website. The original Kaggle dataset has 34 variables which pertain to a variety of socio-demographic, academic, and psychological data.

The integration tools we used ensured that the content of each of the datasets was arranged in a structured format in an Excel sheet while at the same time not interfering with the contents of every dataset. Taking into account that each of the datasets had a different number of feature values and number of instances, the task incorporated a methodical and progressively punctuated connection of the datasets with each other.

To begin with, the irst data set comprised of fifteen features was moved into some of the first columns of the Excel file. Therefore, dataset 2 was appended, beginning with another set of columns in order to prevent overlapping of the previous columns of the first data set. Lastly, dataset 3 was progressed in such a way that its elements and instances kept carrying on where dataset 2 had finally stopped.

This method of merging has given rise to an extensive socialCops data but many missing values have also been introduced, since the natural features of the two other datasets differed by a large margin. We had to find a way to deal with such missing values and make all the datasets compatible so specific imputation techniques had to be used. Missing values have been filled in such a way that the mean of the similar values is used within the columns for numerical features. This technique ascertained that the statistics of the numerical information did not get distorted. However, as for categorical features, we employed the mode in the respective columns in order to keep the category ratios intrinsic to the data set.

Taking this into consideration, the following steps or rather changes enabled us to get a consolidated, strong data that in return increased the chances of building effective models capable of predicting and understanding student performance. With such precision in merging and pre-processing, the resulting data set was a wide set of data with minimal shortcomings which hugely improved the reliability of the study.

The primary attributes include gender (male or female), age, and institution type (government or non-government). IT Student (whether the student is currently enrolled in an IT program: Yes/No), *etc...* Figure 2 depicts the dataset that specifically emphasizes the academic achievements of college students.

The dataset also contains information on the duration of the class attended, whether the student uses the Self (LMS) for self-study, the number of resources visited by the student, the number, the number of discussion posts made, the number of days the student was absent, the student's class level, and the student's final grade, *etc.*

Further, the adaptivity level can be regarded as low, medium, or high. It also features the emotional state, which can be divided into happy or sad, and the focus state, which can either be focused or unfocused.

To facilitate predictive modeling, the features in this article have been classified into five distinct class labels: Performance Level (Low, Medium, High), Final Grade (Pass, Fail), Adaptivity Level (Low, Medium, High), Emotional State (Happy, Sad), and Focus Status (Focused, Unfocused). The analysis and results sections of this article will refer to these

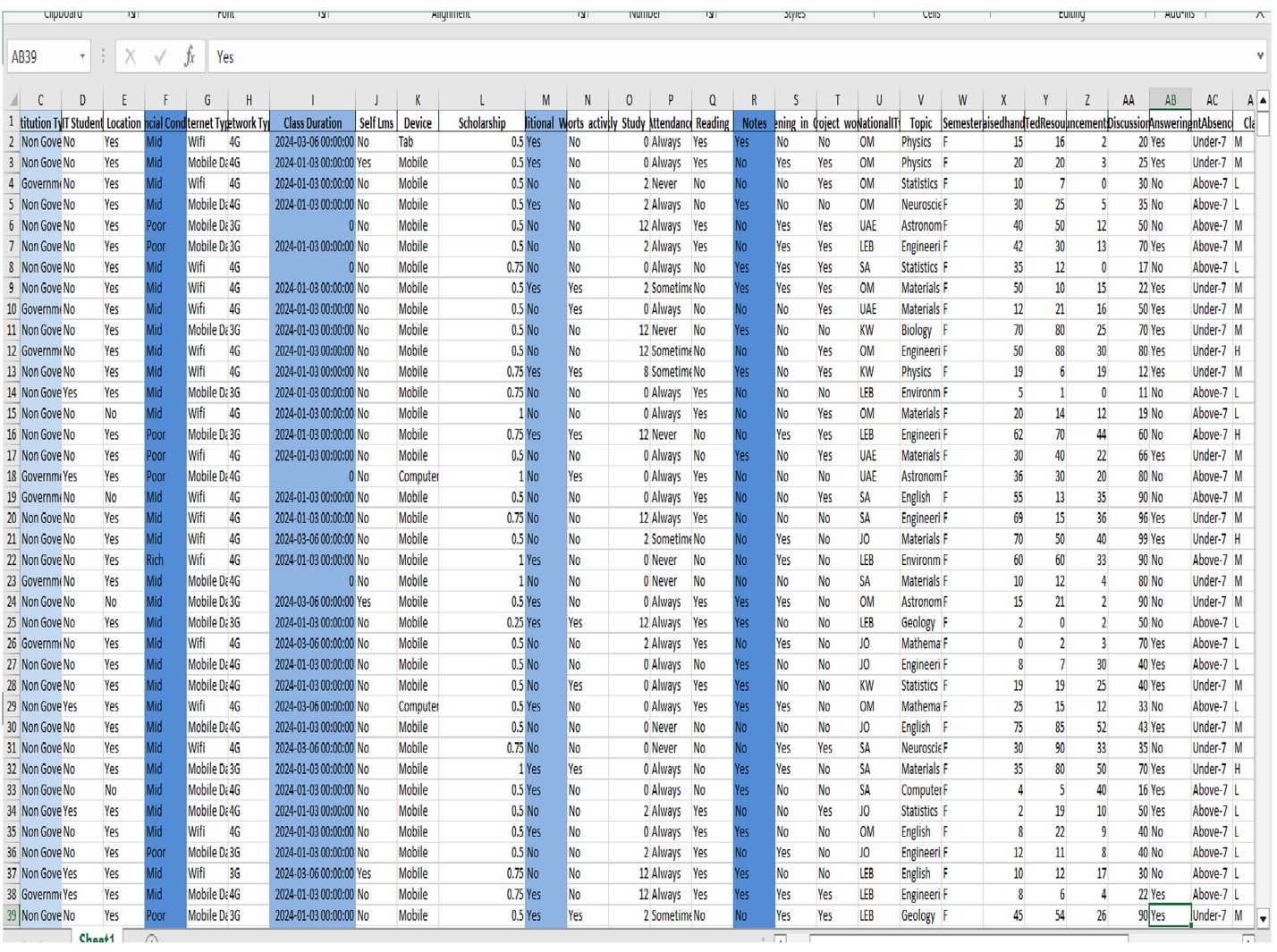

**Figure 2** The dataset focuses on the academic accomplishments of college students.

class labels as Class One, Class Two, Class Three, Class Four, and Class Five, respectively. This extensive dataset allows for the utilization of sophisticated machine learning and deep learning algorithms to forecast and improve different facets of student performance.

## Preprocessing methods

Once the dataset was gathered, we utilized various preprocessing techniques, employing advanced artificial intelligence methods, to ensure its suitability for the prediction process. The following subsections give details of these steps (Fig. 3).

These methods are crucial for enhancing the quality and precision of the predictions. The preprocessing stages encompass the following:

1) Performing missing values check: This stage entails detecting any instances of missing data in the dataset. Missing values can have a substantial impact on the performance of

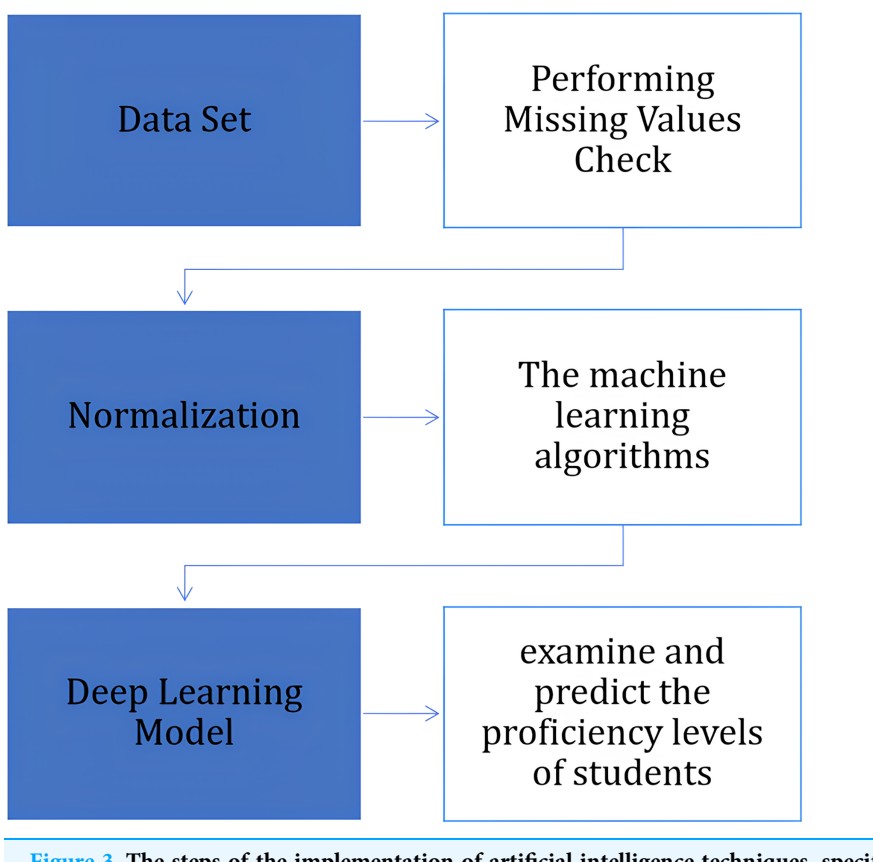

**Figure 3** **The steps of the implementation of artificial intelligence techniques, specifically machine learning and deep learning.**

machine learning models; therefore, they were appropriately addressed through imputation or removal (*Ren et al., 2023*).

2) In order to convert categorical attributes into numerical values, the LabelEncoder methodology was adopted. In this process, one can go through each of the categorical variables and map the categories to corresponding numerical values which can start from 0. Such an approach is very common in cases when numerical values are necessary to provide machine learning results (*Bai, Kong & Gomes, 2022*).

3) Normalization: It was also considered how to change the range for all the features. All the features were scaled using the MinMaxScaler method, which allows rescaling the features to the maternal metric within the range 0 to 1 which in turn speeds up the process of the minimization of the gradient and at the same time guarantees that all the features are considered as having equal weights in AI methods (*Huang et al., 2023*). Thus, normalizing data improves how fast models converge as well as the accuracy of their predictions by minimizing the effect of anomalies on them.

4) Feature extraction: Apart from normalizing, feature extraction techniques are likewise applied at identifying selective features from the dataset which are most relevant. By mimicking anomaly detection process, this will lower problem dimensionality and concentrate only at those attributes that are more significant to predict outcomes.

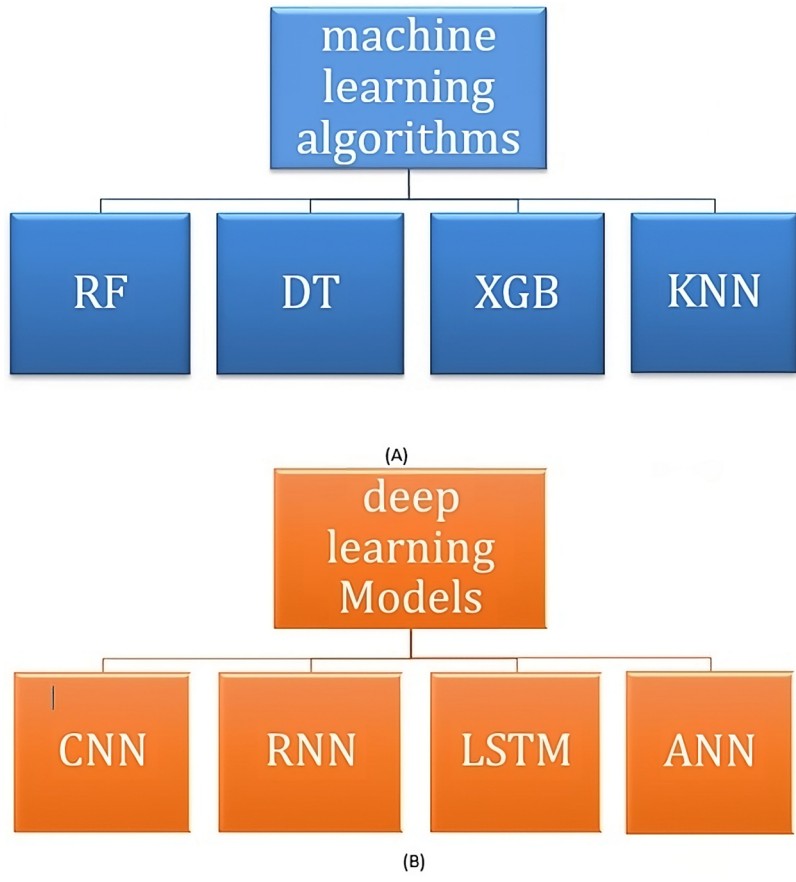

**Figure 4 Classes of machine learning and deep learning.**

5) In process of finishing all pre-processes, we used different techniques of Artificial Intelligence to project many outputs concerned with students, including the final grade, adaptive level, emotional states Happy or Sad and focus status Focused or Focusedless within the class. Machine learning models use include RF, DT, XGB and KNN as depicted in Fig. 4. The selection of these algorithms was based on their resilience and efficiency in managing diverse datasets. In addition, four deep learning models to improve the accuracy of predictions are CNN, RNN, LSTM, and ANN.

## Data reliability overview

Every step in the data preprocessing plays a crucial role in ensuring data quality, contributing to the increased reliability of the models employed. By addressing missing values, encoding categorical variables, normalizing data, and extracting features, we ensure that the data used is accurate and comprehensive. This, in turn, leads to more reliable results in the analyses. Ensuring data reliability is essential for drawing valid conclusions from our research, as the quality of predictions is inherently linked to the integrity of the input data.

A brief description of each algorithm is as follows:

1. Random forest

Machine learning has a technique referred to as the Random Forest methodology, which is prominent and falls under the category of supervised learning methods. In classification and regression problems, it is capable of handling both categories which are different regarding their chronology, meaning that they occur at separate and distinct times during development or processing stages (of the machine). Ensemble learning enhances its performance by solving difficult problems using multiple classifiers such as this one. Decision trees form the building blocks of the random forest classifiers; they are trained on different subsets of a dataset. There are multiple variations of decisions which come as a result from many trees hence raising its prediction accuracy most commonly done by averaging them together. The forest shows higher accuracy when different trees are added but the problem of overfitting is diminished hence maintaining its consistency in doing so over several measurements without such worries creeping upon us anymore following the introduction additional trees into it. Essentially, random forest predicts accurately through the use of many estimators working separately in decision trees which are then combined together (*Zhang et al., 2020b*; *Sheykhmousa et al., 2020*).

2. Decision tree

Decision trees are supervised machine-learning approaches used for regression and classification. They function by learning a series of nested if-else questions based on the data's properties and then predicting the outcome. The objective is to build a model that predicts the value of a target variable by applying fundamental decision rules derived from data attributes. The Recursive Binary Splitting algorithm splits the data iteratively up to a certain extent such as when information gain is the highest or when impurity is the least. The split condition extends until a specified limit such as the maximum depth of the tree or when the model experiences no further performance improvements. Decision trees that are not properly pruned are at risk of overfitting (*Costa & Pedreira, 2023*).

3. eXtreme gradient boosting

Extreme gradient boosting, XGBoost in short, is an optimized and distributed gradient boosting library which is appreciated for its efficiency, flexibility and easy migration to a new notebook. It consists of gradient-boosted decision trees geared towards quick and efficient computation. XGBoost will typically consist of a process where multiple decision trees are constructed in an iterative fashion and then improved by adjusting them to a certain loss at every stage. It is this loop of such improvement that leads to XGBoost being recognized for speed and performance accuracy making it highly desirable in data science competitions and many other areas (*Qiu et al., 2022*).

4. K-nearest neighbors

KNN algorithm can be viewed as a conceptually easy to understand method for both classification and regression. This algorithm is based on the hypothesis that there exists more than likely a relationship between the value of an observed data point and the value of a data point with in the same class, if the two have similar set of features. In this regard, classification means that upon introducing a new data point devoid of any class, KNN

determines the k-most similar data points with class information and uses the data points to make a prediction. The class of the offspring is decided here by picking the most frequent class out of the neighbors of this query point for classification, but by taking the average in case of regression. KNN, after receiving a test instance, calculates all the lengths from the nearest distance to all training instances and determines all the values of neighbors2...neighborsK. We assign the class label to the query instance based on the majority vote or average for regression over the k-nearest neighbors in this process (*Zhang, 2021*).

5. Convolutional neural network

CNNs are a type of neural network specifically built to interpret structured grid data like photographs. CNNs have grown in popularity due to their ability to learn spatial hierarchies of features from input data in an automated and adaptive manner. Convolutional, pooling, and fully connected neural network layers represent CNN architecture's essential elements. Operationas of convolutional layers involve a sequence of filters that scan the input image to detect such high-level features as edges, backgrounds, orientated patterns, *etc.* Pooling layers are attached to the output of Conv layer like a small mask. These layers essentially reduce the dimensions of the input volume, making it what is called a Lower volume, and therefore, it helps to reduce parameters in the network as well as compute less volume of data. All these aspects of the spatial-temporal dynamics are used in representative fashion by fully connected neural network layers which are towards the end of the network to perform one's final classification or regression probabilities (*Kattenborn et al., 2021*).

6. Recurrent neural network

Sequential data structures such as recurrent neural networks are best suited for such data as a result of its temporal nature requiring that the data points be arranged in a specific sequence. In doing so, these RNNs embody memory of past sequence elements by utilizing a pre-determined hidden state including a capacity to adjust for the contextual dependencies of the sequences. Recent advancements in RNNs have concentrated on tackling challenges such as vanishing and bursting gradients, which can impede learning in extended sequences. Techniques include the use of gating mechanisms, which have broadened the application of RNNs to complicated sequence problems such as speech recognition, language modeling, and even real-time forecasting in the financial market (*Sherstinsky, 2020*).

7. Long short-term memory

For learning purposes, it is often the case that these sequences are maintained in long term memory at an extent; LSTM stands as a mechanism for maintaining such long sequences within networks, and it is possible to illustrate its involvement in this over a long period from the use of recurrent networks. LSTMs accomplish this by addressing the gradient-plane problem of recurrent neural networks whereby the vanishing gradient problem due to the extended durations of the sequence is tackled by having a persistent cell. The design of the LSTM model has the information hidden in three elements: an input gate, a forget gate, and an output gate. These serve a very crucial role of enabling the model to learn information that is critical and keep it until later time hence making LSTMs very

useful in tasks such as language modeling, machine translation or sentiment analysis (*Landi et al., 2021*).

8. Artificial neural network

To be less jargonistic, ANN incorporate sheets of nodes, also known as neurons, which are connected with an interconnection called as a weight. Generally, ANNs have input, output, and layer/s as shown in the picture. There is a function called "activation function" inside the cell which is useful in calculating the difference (sum) of the inputs and the outputs, you can also use ReLU Activation Function or Sigmoidal Activation Function. The latest achievements in the ANNs field involve the development of more improved and challenging systems like that of Deep Residual Networks (ResNets) as well as the development of transfer learning as a technique that aids in calibrating a pre-adapted ANN model for a particular application. These recent developments have further enhanced the application of ANN tools for such tasks as image recognition, speech–language analysis and greatly solving complex puzzles (*Kurani et al., 2023*).

This research utilizes the design science research (DSR) to create a framework based on AI that connects CIFAR convolutional neural networks with random forest inorder to optimize e-learning environments.

## Algorithm selection and rationale

The Random Forest algorithm was selected because of its ability to effectively and efficiently reduce overfitting while simultaneously managing large datasets that contain a large number of variables. CNN was utilized in order to conduct the analysis of patterns in student interactions and the identification of behavioral trends. CNN's strength in the extraction of complex features was capitalized on to achieve this. They were chosen so as to ensure that accurate predictions and useful insights that can enhance the level of interactivity and productivity among individuals are made.

## EXPERIMENTAL RESULTS

This part demonstrates the results of applying machine learning algorithms (RF, DT, XGB, KNN) and deep learning models (CNN, RNN, LSTM, ANN) in predicting five different class labels: Performance Level, Final Grade, Adaptivity Level, Happy *vs* Sad, and Focused *vs* Unfocused. Their accuracy values were used to determine how good they are in predicting student outcomes. Variables in the dataset include academic performance, engagement levels, and learning preferences which were employed to show the effectiveness of the framework in enhancing student outcomes. Equation (1) is used to calculate this accuracy value where TP stands for true positive, TN stands for true negative, FP stands for false positive while TN is indeed negative.

$$\text{Accuracy} = \frac{\text{TP} + \text{TN}}{\text{TP} + \text{TN} + \text{FP} + \text{FN}} \tag{1}$$

## Machine learning results

Table 1 shows the machine learning results for each class label in the dataset mentioned. In class 1, the DT gave the best accuracy results compared with the others of 95%. In class 2,

**Table 1 Machine learning results—accuracy.**

|       | Class One | Class Two | Class Three | Class Four | Class Five |
|-------|-----------|-----------|-------------|------------|------------|
| KNN   | 0.90      | 0.92      | 0.88        | 0.96       | 0.97       |
| DT    | 0.95      | 0.94      | 0.87        | 0.97       | 0.98       |
| RF    | 0.94      | 0.95      | 0.86        | 0.95       | 0.96       |
| XGB   | 0.91      | 0.94      | 0.84        | 0.955      | 0.965      |

the RF gave the best accuracy results compared with the others at 95%. In class 3, the KNN gave the best accuracy results compared with the others of 88%. In classes 4 and 5, the DT gave the best accuracy results compared with the others at 97% and 98%, respectively.

Table 2 presents the pairwise relationship between each of the two classes in the dataset in the KNN. Where examining the connections between class labels offers substantial scientific and academic worth by uncovering the associations and interconnections among various variables in the dataset. This analysis deepens the comprehension of the interplay between different factors, such as Performance Level and Adaptivity Level, which can enhance the precision and resilience of predictive models. Furthermore, it assists in recognizing significant influences that impact student achievements, providing valuable insights for implementing focused educational approaches to address particular requirements. Pairwise relationship analysis provides a more comprehensive understanding of the intricate dynamics of educational data, which in turn helps create more impactful learning interventions and improve student achievement. The highest value is between class 2 and class 2, with 0.901. at the same time, the lowest value is between class 5 and class 5 with 0.050.

Table 3 presents the pairwise relationship between each of the two classes in the dataset in the DT. The highest value is between class 1 and class 1, with 0.983. At the same time, the lowest value is between class 5 and class 2, which is 0.003.

Table 4 presents the pairwise relationship between each of the two classes in the dataset in the RF. The highest value is between class 1 and class 3, with 0.974. at the same time, the lowest value is between class 3 and class 2 with 0.015.

Table 5 presents the pairwise relationship between each of the two classes in the dataset in the XGB. The highest value is between class 2 and class 2, with 0.996. at the same time, the lowest value is between class 2 and class 1 with 0.056.

## Deep learning results

The machine learning predictions of each class from the data set mentioned are depicted in Table 6. The figures for class 1 indicates that the CNN had the highest level of accuracy at 97% leaving behind all other classes. The LSTM came the closest to out of all the methods yet the accuracy rate in class 2 gave the highest accuracy of 96%. In class 3 the other category attaining the maximum accuracy with valie of 89% was LSTM. In case of classes 4 and 5 both RNN provided the highest accuracy of 98%.

**Table 2  KNN pairwise relationships.**

| Class | One | Two | Three | Four | Five |
|---|---|---|---|---|---|
| One | 0.741 | 0.789 | 0.829 | 0.557 | 0.544 |
| Two | 0.588 | 0.901 | 0.238 | 0.418 | 0.855 |
| Three | 0.260 | 0.420 | 0.714 | 0.973 | 0.493 |
| Four | 0.157 | 0.630 | 0.422 | 0.251 | 0.874 |
| Five | 0.425 | 0.885 | 0.484 | 0.218 | 0.050 |

**Table 3  DT pairwise relationships.**

| Class | One | Two | Three | Four | Five |
|---|---|---|---|---|---|
| One | 0.983 | 0.305 | 0.288 | 0.762 | 0.482 |
| Two | 0.731 | 0.790 | 0.395 | 0.084 | 0.906 |
| Three | 0.642 | 0.204 | 0.612 | 0.900 | 0.399 |
| Four | 0.529 | 0.846 | 0.684 | 0.677 | 0.874 |
| Five | 0.522 | 0.003 | 0.562 | 0.518 | 0.233 |

**Table 4  RF pairwise relationships.**

| Class | One | Two | Three | Four | Five |
|---|---|---|---|---|---|
| One | 0.127 | 0.743 | 0.974 | 0.941 | 0.338 |
| Two | 0.194 | 0.653 | 0.619 | 0.585 | 0.370 |
| Three | 0.680 | 0.015 | 0.810 | 0.681 | 0.256 |
| Four | 0.722 | 0.121 | 0.773 | 0.477 | 0.391 |
| Five | 0.905 | 0.409 | 0.324 | 0.884 | 0.486 |

**Table 5  XGB pairwise relationships.**

| Class | One | Two | Three | Four | Five |
|---|---|---|---|---|---|
| One | 0.548 | 0.659 | 0.940 | 0.301 | 0.100 |
| Two | 0.056 | 0.996 | 0.646 | 0.487 | 0.803 |
| Three | 0.716 | 0.905 | 0.250 | 0.426 | 0.067 |
| Four | 0.194 | 0.170 | 0.269 | 0.286 | 0.243 |
| Five | 0.880 | 0.574 | 0.194 | 0.284 | 0.419 |

Table 7 presents the pairwise relationship between each of the two classes in the CNN dataset. The highest value is between class 2 and class 2, with 0.90. At the same time, the lowest value is between class 3 and class 5, which is 0.159.

**Table 6 Deep learning results—accuracy.**

| Class/ML | One | Two | Three | Four | Five |
|---|---|---|---|---|---|
| CNN | 0.92 | 0.93 | 0.84 | 0.98 | 0.98 |
| RNN | 0.97 | 0.95 | 0.82 | 0.98 | 0.985 |
| LSTM | 0.95 | 0.96 | 0.89 | 0.94 | 0.90 |
| ANN | 0.92 | 0.91 | 0.80 | 0.925 | 0.91 |

**Table 7 CNN pairwise relationships.**

| Class | One | Two | Three | Four | Five |
|---|---|---|---|---|---|
| One | 0.070 | 0.901 | 0.802 | 0.849 | 0.432 |
| Two | 0.324 | 0.261 | 0.201 | 0.787 | 0.445 |
| Three | 0.709 | 0.476 | 0.298 | 0.504 | 0.159 |
| Four | 0.858 | 0.882 | 0.276 | 0.836 | 0.411 |
| Five | 0.485 | 0.534 | 0.488 | 0.787 | 0.293 |

**Table 8 LSTM pairwise relationships.**

| Class | One | Two | Three | Four | Five |
|---|---|---|---|---|---|
| One | 0.930 | 0.884 | 0.210 | 0.990 | 0.978 |
| Two | 0.889 | 0.616 | 0.755 | 0.794 | 0.353 |
| Three | 0.500 | 0.745 | 0.005 | 0.108 | 0.059 |
| Four | 0.372 | 0.190 | 0.553 | 0.133 | 0.204 |
| Five | 0.289 | 0.898 | 0.203 | 0.077 | 0.569 |

**Table 9 RNN pairwise relationships.**

| Class | One | Two | Three | Four | Five |
|---|---|---|---|---|---|
| One | 0.494 | 0.571 | 0.758 | 0.788 | 0.738 |
| Two | 0.751 | 0.243 | 0.091 | 0.556 | 0.202 |
| Three | 0.872 | 0.091 | 0.161 | 0.597 | 0.769 |
| Four | 0.033 | 0.105 | 0.392 | 0.616 | 0.514 |
| Five | 0.476 | 0.056 | 0.072 | 0.833 | 0.907 |

Table 8 presents the pairwise relationship between each of the two classes in the dataset in the LSTM. The highest value is between class 1 and class 4, with 0.990. At the same time, the lowest value is between class 5 and class 4, with 0.077.

Table 9 presents the pairwise relationship between each of the two classes in the dataset in the RNN. The highest value is between class 5 and class 5, with 0.907. At the same time, the lowest value is between class 5 and class 2 with 0.056.

**Table 10 ANN pairwise relationships.**

| Class | One | Two | Three | Four | Five |
|---|---|---|---|---|---|
| One | 0.955 | 0.142 | 0.061 | 0.657 | 0.779 |
| Two | 0.996 | 0.836 | 0.505 | 0.214 | 0.945 |
| Three | 0.616 | 0.409 | 0.773 | 0.578 | 0.616 |
| Four | 0.650 | 0.155 | 0.220 | 0.572 | 0.649 |
| Five | 0.410 | 0.993 | 0.700 | 0.840 | 0.319 |

Table 10 presents the pairwise relationship between each of the two classes in the dataset in the ANN. The highest value is between class 2 and class 1, which is 0.996. At the same time, the lowest value is between class 1 and class 3, with 0.061.

As the results, the deep learning algorithms outperformed the machine learning algorithms based on accuracy values in all classes.

We tested the suggested structure by considering the main metrics—prediction precision, level of involvement, and computational effectiveness—it should meet educational as well as technical necessities.

## Test for performance evaluation

A t-test was administered to quantify the deviations in performance observed across various machine learning and deep learning models. This assessment consists of seeing up 14 random comparisons. The results of the t-test are shown in the table above. Moreover, the following bar chart shows the t-statistic for each comparison, where a t-statistic is the absolute value of the difference in performance between each pair of models. All $p$-values exceed to 0.05, which means there isn't any statistically significant difference in performance of any two models in this dataset. Hence, the performance of these models on this dataset is similar, and there is no difference in the selection of the models presented in Table 11 (Fig. 5).

This is consistent with the findings of previous studies in the literature such as those made by *Mayer (2022)* regarding cognitive learning theories. Consequently, the application of Artificial Intelligence models like CNN as well as RandomForest witnessed marked improvement in students' grades due to support of associative learning methods (*Bhat, Mangardich & Sabbagh, 2022*). On a similar note, situational learning theory underscores how learners need to interact when they're involved in various activities in a computerized environment that combines both virtual and physical settings together (blended learning) (*Potvin & Hasni, 2014*; *Lai, 2021*). These results therefore bridge a critical gap between literature mixing AI technologies and blended learning models.

## Algorithm performance comparison

In Table 12 algorithms such as RF and CNN are compared in this table. The following primary metrics are used to gauge performance; accuracy, recall (*i.e.*, sensitivity), and precision.

**Table 11  T-test performance evaluation.**

| Model 1 | Model 2 | T-Statistic | *P*-Value |
|---|---|---|---|
| KNN | DT | −0.61813 | 0.553673 |
| KNN | RF | −0.23905 | 0.81708 |
| KNN | XGB | 0.141201 | 0.891202 |
| KNN | CNN | −0.12937 | 0.900259 |
| KNN | RNN | −0.42477 | 0.682195 |
| KNN | LSTM | −0.09035 | 0.93023 |
| KNN | ANN | 1.135235 | 0.289142 |
| DT | RF | 0.375823 | 0.716819 |
| DT | XGB | 0.674008 | 0.519302 |
| DT | CNN | 0.373182 | 0.718707 |
| DT | RNN | 0.027472 | 0.978756 |
| DT | LSTM | 0.587427 | 0.573119 |
| DT | ANN | 1.612855 | 0.145441 |
| RF | XGB | 0.34493 | 0.73904 |
| RF | CNN | 0.063436 | 0.950976 |
| RF | RNN | −0.25107 | 0.80809 |
| RF | LSTM | 0.174078 | 0.866129 |
| RF | ANN | 1.312454 | 0.225768 |
| XGB | CNN | −0.23423 | 0.820688 |
| XGB | RNN | −0.49768 | 0.6321 |
| XGB | LSTM | −0.2267 | 0.826344 |
| XGB | ANN | 0.892625 | 0.398112 |
| CNN | RNN | −0.27406 | 0.790982 |
| CNN | LSTM | 0.068439 | 0.947116 |
| CNN | ANN | 1.064115 | 0.318338 |
| RNN | LSTM | 0.384185 | 0.710851 |
| RNN | ANN | 1.239355 | 0.250338 |
| LSTM | ANN | 1.284025 | 0.235069 |

Specifically, the Random Forest algorithm had an accuracy level of 92.5%. This meant that it correctly classified 92.5% of instances. Further, the algorithm's ability to identify positive incidents correctly is its recall, which stands at 91.0%. It may be understood by examining the algorithm's precision the number of truly positive cases the model identifies as such. As a matter of fact, the random forest had a precision of 93.2%.

On the flip side, in terms of accuracy, CNN was just slightly better at percentage 95.1%. Also CNN's recall is 94.5% which means when looking at positive instances only compared to RF that has only 91.0%. In the same context, its precision rates are above those of RF at 95% which further shows that CNN accurately identifies true positives. In all aspects, therefore, CNN is better than RF meaning it is more accurate and reliable on this particular task.

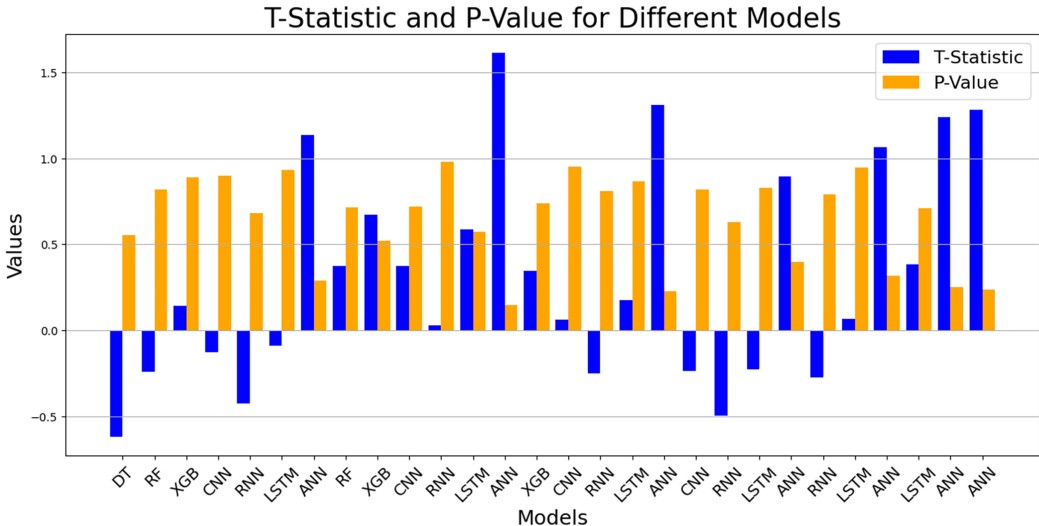

**Figure 5** T-test performance evaluation.           

**Table 12 Algorithm performance comparison.**

| Algorithm | Accuracy (%) | Recall | Precision |
|---|---|---|---|
| Random Forest (RF) | 92.5 | 91.0 | 93.2 |
| CNN | 95.1 | 94.5 | 95.0 |

## DISCUSSION

First and foremost, a comprehensive overview on how AI has got a remarkable capability to come up with creative and adaptive online learning systems that greatly improve learning outcomes will be given in this article. Education programs have personalized learning modes which will be achieved through artificial intelligence in this way it becomes possible for design and development of dynamic adaptive online learning systems by virtue of AI which have the extraordinary ability of improving academic performance considerably.

One most striking thing about this advanced research outcome is how CNN has consistently been outstanding in predictive tasks across all phases. In comparison with other methods such as deep learning techniques or traditional machine supervision mechanisms; it is noticeable that its higher accuracy rate ensures that it remains a formidable tool within the field of AI for instance at present time (cite). Moreover, the dramatic growth in CNN reflects not just this but also general change concerning artificial intelligence over time. These developments have significant consequences for the tomorrow's schools where we see an AI that keeps moving frontiers to offer smarter learning spaces that will be more responsive to individual needs.

# CONCLUSION

Ability to implement AI technologies in e-learning systems has been brought out redefined by this study where supervised monitoring of student interactions in a traditional Middle Eastern educational environment has been done. Furthermore, much use of different AI methodologies were made to provide both student and teacher support. We have appreciably demonstrated the advantages for which can be reached depending on the technologies present in the system.

This was extensively operative in that, it hit the ground running into the matters of Research Questions 1 (RQ1) and 2 (RQ2) pretty well. Concerning RQ1, it is observed that performance of students has an improvement and issues of retention and accessibility are effectively dealt with; these were the practices that were put into action. This is because we were able to apply AI procedures on an extensive data set and were able to predict different performance metrics such as the final grade, adaptivity, emotions, and attentiveness levels of the student. This function has been a drive, resulting in the enhancement of student engagement and consequently academic productivity.

As in RQ2, e-learning performance was within meaningfully proposed scope with the integration of AI features in the e-learning systems. Random Forest, Decision Tree, XGBoost, K-nearest neighbors and other machine learning models, as well as convolutional neural networks, recurrent neural networks, long-short term memory networks, and artificial neural networks as deep learning models, also showed potent abilities in predictions. This data showed that deep learning models always gave more accurate results than traditional machine learning models.

The current findings are clear that modern resources should be injected into providing adaptable, convenient and effective E learning programs. With artificial intelligence, training facilitators get live data concerning how students are doing and this prompts them to tackle any problems that could be affecting the students promptly and specifically. This enhances understanding and performing well in academics while at the same allows students to adopt self-paced learning terminated in enhanced knowledge acquisition.

This research contributes to the growing body of knowledge by demonstrating the effectiveness of DSR in optimizing e-learning environments through AI techniques. The proposed framework improves student outcomes and offers a scalable solution for future implementations.

## Limitations and future work

Future research will be placed on making the dataset more comprehensive to include different types of students from different regions, in order to enable broader applications of the models in question. Real time data obtained from biometrica by using wearable technology can enhance the precision of such predictions.

Another more effective strategy is probably to reproduce data gathered by managing IoT technologies in the environment that contains the most Middle Eastern students since the results of that research was more encouraging polishing the achievements of the student.

Finally, AI and IoT technologies should be positioned into the voice of technology and the acceptance and use of these technologies in school settings would need quality enabling tools for optimization of resources. This kind of strategy in practising education would be very helpful.

Further research might therefore want to examine how such models apply under different sets of conditions regarding teaching English to young speakers. Moreover, we are yet to know how teachers who use traditional methods can benefit from AI insights that enhance hybrid teaching approaches.

### Funding
The authors received no funding for this work.

### Competing Interests
The authors declare that they have no competing interests.

### Author Contributions
- Rund Mahafdah conceived and designed the experiments, performed the experiments, analyzed the data, performed the computation work, prepared figures and/or tables, and approved the final draft.
- Seifeddine Bouallegue performed the computation work, authored or reviewed drafts of the article, supervision, and approved the final draft.
- Ridha Bouallegue performed the computation work, authored or reviewed drafts of the article, supervision, and approved the final draft.

### Data Availability
The raw measurements are available in the Supplementary Files.

The third-party datasets used in this study are available at:

- Student Performance Prediction Dataset: https://www.kaggle.com/datasets/prajwalkanade/student-performance-prediction-dataset.

- Students' Adaptability Level in Online Education: https://www.kaggle.com/datasets/mdmahmudulhasansuzan/students-adaptability-level-in-online-education.

- xAPI-Edu-Data: https://www.kaggle.com/datasets/aljarah/xAPI-Edu-Data.

### Supplemental Information
Supplemental information for this article can be found online at http://dx.doi.org/10.7717/peerj-cs.2576#supplemental-information.

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
