# Peer review of "Enhancing e-learning through AI: advanced techniques for optimizing student performance"

_PeerJ Computer Science, doi:10.7717/peerj-cs.2576_

## Round 0.1 · original submission · Major Revisions

Dear authors,
You are advised to critically respond to all comments point by point when preparing an updated version of the manuscript and while preparing for the rebuttal letter. Please address all comments/suggestions provided by reviewers, considering that these should be added to the new version of the manuscript.

Kind regards,
PCoelho

Reviewer 1 ·

Basic reporting

This study is well-designed to tackle AI-based framework to monitor and analyze student interactions using advanced algorithms and determining the most efficient methods for blended learning systems. The paper provides recommendations to integrate AI technologies in e-learning systems. Overall, the study's arguments are clear, the structure is logical, the research methodology is sound, and meaningful conclusions are drawn. Find below detailed comments:
• English language fine. No issues detected
• The introduction is well written, and it adequately introduces the subject and shows the motivation and the research questions. However, more studies need to be cited to support arguments. For example, it was indicated that “Many authors proposed that various modern approaches would help with the required improvement” but no citation for these studies. Try to cite three studies at least. Also, go through the introduction (line 79 to 93) there are lots of arguments that need supporting reference.
• The literature is well referenced and relevant. However, I suggest adding 2-3 sentences describing what will be discussed (try to elaborate instead of saying “previous works that are related to this paper in many sections”). Also please give reference for the three theories (Associative, cognitive, and situational theories).
• The discipline differences of the use of technologies in learning such as IOT, sentiment Analysis, AI is unclear, it needs to report the limitation if any at the end of 2.6 section.

Experimental design

• The methodology section was clearly presented. You need to cite the sources for the preprocessing methods used in section 3.2 adequately. Moreover, I suggest adding headings or bullets for the description of each algorithm (* Random Forest (RF), * Decision Tree (DT), ….. etc). Also, I suggest adding a table for all abbreviation in this paper in the appendix.

Validity of the findings

No discussion section that links the literature review with what the study found

Additional comments

• The EXPERIMENTAL RESULTS section should have heading number and subheadings number.
• The conclusions are thoroughly supported by the results presented in the article. However, the study limitations should be discussed in the conclusion

·

Basic reporting

Unambiguous English: The language is professional and clear, though some sentences could be more concise to improve readability.

Introduction & Background: The context is provided, but the background is general. No specific use cases or references are studied. Adding references to related work would strengthen the framing.

Literature Referencing: No explicit references are made to prior work, weakening its scholarly depth. Including relevant literature is recommended. Also, adding key findings from the literature is necessary to understand the current issue in the system to be solved.

Structure: The structure is sound and typical for AI-related abstracts. However, more direct motivation and implications could improve clarity. Few figures are directly generated from AI.

Introduction & Motivation: The subject is introduced well, but the motivation is implied rather than clearly stated. Make the problem clearer upfront.

Formal Results & Definitions: Key terms like "CNN" and "performance metrics" are used but not defined. Brief explanations would help non-expert readers. Also, classes labelled as one, two, three, and four do not suggest anything or the encoding information provided.

Experimental design

Sufficient Discussion: The paper does not explicitly mention data preprocessing. However, given the nature of the research (analyzing student interactions), it is likely that some preprocessing steps were necessary (e.g., data cleaning, normalization, feature extraction). A more detailed discussion would be beneficial to understand the quality and reliability of the data used.

Adequate Description:
The Paper states that the models were evaluated for accuracy, but it lacks specific details about the evaluation methods and metrics used. It also does not mention the model selection process. A more comprehensive explanation of these aspects would be required to assess the validity of the findings.

Insufficient Information:
Based on the Paper alone, it is reviewed that citations are not the latest and not up to the mark.

Validity of the findings

Insufficient Assessment:
The Paper does not explicitly address the impact or novelty of the research. A discussion of how the findings contribute to the existing body of knowledge and their potential implications for the field would be valuable.

Well-Stated Conclusions:
The Paper seems to have well-stated conclusions, but it is difficult to assess their accuracy without more information about the experiments and evaluations.

Unresolved Questions: The paper does not explicitly identify any unresolved questions, limitations, or future directions. This would be a valuable addition to strengthen the research's contribution to the field.

Additional comments

No Comment.

Reviewer 3 ·

Basic reporting

This article examines how Artificial Intelligence (AI) can improve e-learning environments by using advanced predictive algorithms, such as machine learning and deep learning, to optimize student performance.

The study aims to create an AI-based framework for analyzing student interactions and improving learning outcomes.

The results of applying various AI models, such as CNN and RNN, show significant accuracy in predicting performance metrics, with CNN outperforming the other methods.

The introduction discusses the importance of educational systems and the need for modernization, particularly with regard to e-learning. It highlights the growing role of AI in improving learning outcomes, solving accessibility problems, and increasing student engagement.

The document underlines that, despite advances in AI, existing e-learning platforms do not fully exploit its potential, creating a need for frameworks that can better predict and improve student performance. The study aims to bridge this gap by integrating AI to optimize educational outcomes.

The literature study covers several key areas, such as distance learning with cognitive skills, frameworks for e-learning; IoT in learning and artificial intelligence.

However, the document does not present a systematic literature review, nor does it follow an appropriate protocol for this purpose (Prisma 2020, for example).

Experimental design

The methodology follows the stages such as Data collection, Pre-processing, Deep learning models, and AI techniques.

However, it does not follow any proper scientific methodology for this purpose.

An application of Design Science Research (DSR) is suggested with a focus on an artifact, demonstrated and validated.

The sources cited are acceptable, but more scientific criteria are needed to validate and discuss the results.

Validity of the findings

The paper uses a combined dataset that includes student performance prediction dataset student adaptability level in online education, with xAPI-Edu-Data.

The study demonstrates that AI models can predict student performance and improve educational outcomes in e-learning environments.

The dataset is limited to students in the Middle East, which may affect the generalizability of the results to other regions.

The study's dataset is specific to one region, and the authors should recognize the limitations of applying these models to other regions or educational systems.

The study does not provide an exhaustive assessment of how AI systems can be implemented in real-time for continuous feedback during online learning.

Additional comments

The paper focuses on accuracy but does not discuss the computational efficiency of the models. Analyzing the trade-offs between accuracy and computational cost could provide a more balanced view of the methods used.

The results highlight the ability of AI to improve e-learning systems, creating more effective and adaptive learning experiences, which can be of great benefit for personalizing teaching and improving academic performance.

The article is promising and provides important contributions to AI in e-learning.

However, revisions are needed, namely to clarify the presentation of the data, recognize the limitations, and discuss real-world applications (LSR and DSR, as examples).

In the conclusions, they could have presented the main limitations of the study and elaborated on the type of future research to be pursued.

---

## Round 0.2 · accepted · Accept

Dear authors, we are pleased to verify that you meet the reviewer's valuable feedback to improve your research.

Thank you for considering PeerJ Computer Science and submitting your work.

Reviewer 1 ·

Basic reporting

The authors responded to all comments

Experimental design

The methodology is good now

Validity of the findings

The authors responded to all comments

Additional comments

The authors responded to all comments

Reviewer 3 ·

Basic reporting

The changes made to the article follow the recommendations given.

Experimental design

The changes made to the article follow the recommendations given.

Validity of the findings

The changes made to the article follow the recommendations given.

Additional comments

The changes made to the article follow the recommendations given.